# Multispectral Assessment of Sweet Pepper (*Capsicum annuum* L.) Fruit Quality Affected by Calcite Nanoparticles

**DOI:** 10.3390/biom11060832

**Published:** 2021-06-03

**Authors:** Monika Vidak, Boris Lazarević, Marko Petek, Jerko Gunjača, Zlatko Šatović, Ivica Budor, Klaudija Carović-Stanko

**Affiliations:** 1Centre of Excellence for Biodiversity and Molecular Plant Breeding (CoE CroP-BioDiv), Svetošimunska Cesta 25, HR-10000 Zagreb, Croatia; mvidak@agr.hr (M.V.); blazarevic@agr.hr (B.L.); zsatovic@agr.hr (Z.Š.); kcarovic@agr.hr (K.C.-S.); 2University of Zagreb Faculty of Agriculture, Svetošimunska Cesta 25, HR-10000 Zagreb, Croatia; mpetek@agr.hr; 3Agroledina j.d.o.o., Prigorska 32, Moravče, HR-10363 Belovar, Croatia; ivica.budor@gmail.com

**Keywords:** CaCO_3_, fruit quality analysis, image-based analysis, nanotechnology

## Abstract

Sweet pepper (*Capsicum annuum* L.) is one of the most important vegetable crops in the world because of the nutritional value of its fruits and its economic importance. Calcium (Ca) improves the quality of sweet pepper fruits, and the application of calcite nanoparticles in agricultural practice has a positive effect on the morphological, physiological, and physicochemical properties of the whole plant. The objectives of this study were to investigate the effect of commercial calcite nanoparticles on yield, chemical, physical, morphological, and multispectral properties of sweet pepper fruits using a combination of conventional and novel image-based nondestructive methods of fruit quality analysis. In the field trial, two sweet pepper cultivars, i.e., Šorokšari and Kurtovska kapija, were treated with commercial calcite nanoparticles (at a concentration of 3% and 5%, calcite-based foliar fertilizer (positive control), and water (negative control) three times during vegetation). Sweet pepper fruits were harvested at the time of technological and physiological maturity. Significant differences were observed between pepper cultivars as well as between harvests times. In general, application of calcite nanoparticles reduced yield and increased fruit firmness. However, different effects of calcite nanoparticles were observed on almost all properties depending on the cultivar. In Šorokšari, calcite nanoparticles and calcite-based foliar fertilizers significantly increased N, P, K, Mg, Fe, Zn, Mn, and Cu at technological maturity, as well as P, Ca, Mg, Fe, Zn, Mn, Cu, and N at physiological maturity. However, in Kurtovska kapija, the treatments increased only Ca at technological maturity and only P at physiological maturity. The effect of treatments on fruit morphological properties was observed only at the second harvest. In Šorokšari, calcite nanoparticles (3% and 5%) increased the fruit length, minimal circle area, and minimal circle radius, and it decreased the fruit width and convex hull compared to the positive and negative controls, respectively. In Kurtovska kapija, calcite nanoparticles increased the fruit width and convex hull compared to the controls. At physiological maturity, lower anthocyanin and chlorophyll indices were found in Kurtovska kapija in both treatments with calcite nanoparticles, while in Šorokšari, the opposite effects were observed.

## 1. Introduction

Calcium (Ca) has many functions in plants, including fruit formation, development, maturation, and quality [1,2]. Adequate Ca concentration in the plant prevents fruit disorders before and after harvest and increases tolerance to pathogens [3]. Although Ca is one of the most abundant plant nutrients in arable soils, its deficiency usually occurs due to its poor translocation within the plant, and therefore developing tissues require a continuous supply of Ca from the environment [4]. Although foliar fertilization cannot completely replace soil fertilization, it can improve nutrient uptake and efficiency in plants with a large leaf area and it is particularly useful for poorly mobile nutrients [5,6].

Consequently, modern sustainable vegetable production is turning towards methods that mitigate stress events and increase quality and yield; for this reason, foliar fertilizers, biostimulants, and plant growth regulators, as well as recent innovative techniques such as nanotechnology, could play an important role in the improvement of the agricultural products [7,8]. Nanotechnology encompasses many applications in various fields, from agriculture to medicine [9]. From the perspective of sustainable agriculture, nanotechnology has the potential for the development of new innovative types of fertilizers based on the use of slow or controlled release of active compound, such as nanofertilizers, to increase global food production to feed the growing world population [8,10]. Nanoparticle technology is characterized by high reactivity (i.e., larger specific surface area and higher density of reactive regions on the particle surface), therefore nanofertilizers provide more nutrients to the plants than known foliar fertilizers [7,10]. One of the most important properties of nanofertilizers is their ability to penetrate into plants when applied in small amounts as foliar or soil amendments due to their small particle size (<100 nm), and provision of the nutrients with high efficiency and low waste due to their faster and higher translocation to different plant parts [8,11]. The calcite nanoparticles production is achieved by tribomechanical activation (TMA), in which the calcite comminution is not performed by friction and pressure but by the collision of the calcite particles at high speed, so that they are crushed, fragmented, and disintegrated; in this way, the composition of the mineral is not changed—it only breaks into nano- and microparticles with very irregular shape, which significantly increases their active surface area [12]. Tribomechanical treatment causes the micronization of various solid materials, which leads to nanoparticles and changes in the structure and the electric potential of molecules [13]. Today, nanonized/micronized calcite foliar powder is an industrial product produced in large quantities and used in many countries around the world [14]. Due to their positive effects on anatomical, morphological, physiological, physicochemical, and molecular traits, they can also have stronger effects on growth and development, including benefits for fruit production, quality, shelf life, and tolerance to abiotic stress and pest control [11,15,16]. The interaction between the plant and the surfaces of nanoparticles and nanomaterials can positively affect the transport of ions and metabolites and the activity of receptors by changing the microenvironment in terms of energy and charge, and this activity is not dependent on the chemical composition [17]. The nanomaterials are usually sprayed on plants frequently but in different time intervals, as they can increase plant growth if applied in certain concentration ranges (usually in small amounts); for this reason, they are also known as growth stimulants [11,18]. From all this, it could be concluded that the frequent application of nanoparticles has a subtle but ultimately profound impact on the multiple physiological processes at the level of the whole organ or even the whole plant. Therefore, studies aimed at determining the effects of nanoparticles on yield and quality should consider their broad effects on plant metabolism.

In addition, innovative techniques such as color and multispectral imaging and image analysis as the nondestructive quality assessment of fruits in postharvest handling are recognized as valuable tools in achieving sustainable agriculture and in overcoming the shortcomings of suitable conventional techniques [19,20,21]. Since appearance and nutritional quality are the most important factors for consumer acceptance of fruits, there is a great demand for the introduction of new techniques and agricultural inputs to meet these requirements [22]. Thus, development in modern computer techniques have led to fast, cost-effective, and nondestructive approaches to data processing that provide a more efficient way of quality assessment, i.e., one that is reliable and objective [23,24]. At the same time, farmers are looking for sustainable ways to increase productivity and to ensure food safety and quality for consumers, which in turn can result in desirable profits [22].

Sweet pepper (*Capsicum annuum* L.) is one of the most important vegetable crops in the world, not only because of its economic importance but also because of the nutritional value of its fruits [25,26]. There is a great diversity of sweet peppers with regard to their shape, color, or size; their nutritive composition may vary depending on genotype, fruit coloration, developmental stages, growing region, and agricultural practices [26,27,28]. Pepper is an excellent source of various compounds such as essential vitamins (especially vitamin C), minerals, a number of phytochemicals such as carotenoids, capsaicinoids, flavonoids, and tocopherols, natural pigments, among others [25,26,29,30]. The presence of bioactive compounds in sweet pepper makes its products important as functional foods; they have a positive impact on nutritional value for human consumption and may play a role in reducing human microelement deficiencies [25,31].

The objectives of this study are to evaluate the effect of commercial calcite nanoparticles on sweet pepper fruits yield and the chemical, physical, morphological, and multispectral properties of fruits using a combination of the conventional and novel image-based nondestructive fruit quality analysis methods.

## 2. Materials and Methods 

### 2.1. Experimental Design and Treatments

A field trial was carried out at the Maksimir experimental station (45°49′ N; 16°20′ E) of the Department of Seed Science and Technology at the University of Zagreb Faculty of Agriculture, Croatia. The soil (Eutric Cambisol (Siltic)) at the experimental field had a mild acidic pH with moderate nitrogen and sufficient phosphorus and potassium content (Table 1). Two cultivars of sweet pepper *(Capsicum annuum* L.), i.e., Šorokšari (also known as Soroksári) of the bell fruit type and Kurtovska kapija of the capia fruit type, were selected for this experiment due to their common use in either fresh or processed form and also their stable yield based on their good adaptation to the climate conditions in Central and Eastern Europe. Seedlings were grown from commercial seeds in polystyrene seedling trays (52 cm × 32 cm) with 60 pots (4.5 cm in diameter each) in commercial seedling substrate (pH 5.5–6.5, with 1.0 kg NPK 14-10-18 m^−3^ added; heavy metals (mg kg^−1^): Zn 62.43, Cu 19.54, Cd 0.621, Pb 21.22, Mo 0.221, Ni 9.72, Cr 5.87, Hg < 0.01, As 0.0987, Co 6.11), in a greenhouse at a mean daily temperature of 20 °C. Uniformly well-developed (i.e., at stage of 5–6 fully developed leaves’ pairs, 7 weeks old) sweet pepper seedlings were planted in the field on 3 May, 2019, at a 70 cm distance between rows and a 40 cm distance within rows. The single experimental plot consisted of 7 rows with 49 plants per plot. Edge plants were not analyzed due to edge effect. The cultivation was done under conventional agricultural practices with single fertilization at the beginning of the vegetation (150 kg ha^−1^ NPK 7-20-30). The climatic conditions during the growing season are presented in Figure 1.

The field trial was set up as a randomized complete block design with three replicates. The commercial calcite nanoparticles Eco Green (Agroledina j.d.o.o.) were applied, according to producers’ guidelines, three times during vegetation, while Zeogreen+P (Velebit Agro d.o.o.), a calcite-based foliar fertilizer (CaCO_3_, 69.80%; Na_2_O, 0.25%; Cu, 7.73 mg kg^−1^; SiO_2_, 14.84%; N, 0.132%) was used as positive control, and plain water represented negative control. The first application was at the beginning of fruit formation (2 July 2019), followed by two foliar sprayings in intervals of 7 days (9 July 2019, and 16 July 2019). The calcite nanoparticles, calcite-based foliar fertilizer, and water were sprayed until complete wetness of the plants. The treatments with the treatment abbreviation, calcite and calcium content in the fertilizer products and concentration applied are given in Table 2.

The first harvest was performed at technological fruit maturity, which occurred seven days after the third foliar treatment (23 July 2019). The second harvest was performed in the phase of physiological fruit maturity (3 September 2019) using the remaining plants.

### 2.2. Yield and Yield Components

While all the remaining traits were scored using the fruits collected in both harvests, the data on yield and yield components were scored at second harvest only. Fruit yield per plot was recalculated to t ha^−1^, while the number of fruits per plant and the fruit weight (g) were calculated as the average of all fruits per plot.

### 2.3. Chemical Properties of the Fruit

The sugar content, total organic acids’ content, and pH value were determined in the juice squeezed from the pericarp (without placental tissue). Sugar content (% Brix) was determined using a digital refractometer (Atago, PR-101; The Front Tower Shiba Koen, 23rd Floor, 2-6-3 Shiba-koen, Minato-ku, Tokyo 105-0011, Japan), while total organic acids’ content was determined by the titration method using 1 M NaOH [36] and expressed as g tartaric acid L^−1^. Juice pH was determined electrometrically using a pH meter with the combined electrode (Mettler Toledo, FE20/EL20; Im Langacher 44, 8606 Greifensee, Switzerland).

The mineral composition of sweet pepper fruits was determined in the edible part of the fruit (pericarp only, without placental tissue). The pericarp was cut into smaller pieces in order to make a homogeneous sample. After digestion, phosphorus (P) was determined using a spectrophotometer (Thermo Fisher Scientific, Evolution 60 S; Thermo Fisher Scientific, 168 Third Avenue, Waltham, MA USA 02451), potassium (K) using a flame photometer (Jenway, PFP-7; Cole-Parmer, Beacon Road, Stone, Staffordshire, ST15 OSA, United Kingdom), while calcium (Ca), magnesium (Mg), iron (Fe), zinc (Zn), manganese (Mn) and copper (Cu) were determined using an atomic absorption spectrometer (AAS Solar, Thermo Scientific; (former Unicam), Cambridge, England) [32]. Total nitrogen (N) content was determined using the modified Kjeldahl method [37].

### 2.4. Physical and Morphological Measurements of the Fruit

Fruit morphological parameters, i.e., length (mm), width (mm), convex hull, minimal circle radius (mm), minimal circle area (mm^2^), and minimal rectangle area (mm^2^), were scored from color images using the DA^TM^ analysis software (PhenoVation B.V., Wageningen, The Netherlands). After imaging, the same fruits were used to measure pericarp firmness and thickness. Pericarp firmness (kg cm^−2^) was measured using a penetrometer (PCE-PTR 200, PCE Deutschland GmbH, Im Langel 4, 59872 Meschede, Germany) with a 6 mm diameter probe, and pericarp thickness (mm) using a digital caliper, both at the fruit equatorial zone at the diametrically opposite sides.

### 2.5. Multispectral Analysis of the Fruit

Fruit color and multispectral analysis were performed using CropReporter^TM^ multispectral imaging chamber and the DA^TM^ analysis software (PhenoVation B.V., Wageningen, The Netherlands). All images were captured with the 10 MP lens, 200 LP mm^−1^ resolution, 400–1000 nm spectral range, and a 1.3 MP, 1296 × 966 pixels CCD camera. Five fruits per field plot were imaged from a 40 cm distance under 250 µmol m^−2^ s^−1^ broadband white LEDs (3000 K) illumination. Reflectance images were captured at the following wavelengths: blue at 475 nm, green at 550 nm, red at 640 nm, specific green (SpcGrn) at 510–590 nm, chlorophyll at 730 nm, anthocyanin at 540 nm, near infra-red (NIR) at 769 nm, and far red at 710 nm. From captured reflectance images, raw reflectance parameters were analyzed, and the chlorophyll index (CHI) and anthocyanin index (ARI) were calculated. Chlorophyll index was calculated as CHI = Reflectance (730)^−1^ − Reflectance (769)^−1^ [38], and whereas anthocyanin index was calculated as ARI = Reflectance (540)^−1^ − Reflectance (710)^−1^ [39]. The hue (HUE), saturation (SAT), and value (VAL) were calculated after converting red, green, and blue into values between 0 and 1.

Hue (0–360°) was calculated as follows:

HUE = 60 × [0 + (Green − Blue)/(max − min)], if max = Red;

HUE = 60 × [2 + (Blue − Red)/(max − min)], if max = Green;

HUE = 60 × [4 + (Red − Green)/(max − min)], if max = Blue.

In the case of HUE < 0, 360 was added.

The value (0–1) was calculated as VAL = (max + min)/2, while the max and min were selected from the red, green, and blue. Saturation (0–1) was calculated as SAT = (max − min)/(max + min) if VAL > 0.5, or SAT = (max − min)/(2.0 − max − min) if VAL < 0.5, while max and min were selected from the red, green, and blue.

### 2.6. Statistical Data Analysis

The analyses of data on yield and yield components were carried out using the model based on the actual design (RCBD), thus including the effects of cultivars (*n* = 2), treatments (*n* = 4) and their interaction, as well as the effect of replicates. The partitioned F-tests were performed using the SLICE statement to examine the significance of treatments within cultivars, followed by Tukey’s HDS test. This model was extended to fit the data collected from both harvests, for all the remaining traits. It was done by considering two harvests as subplots of the original plot, which therefore means using the split-plot-in-time mixed model. Cultivars, treatments, and their interaction, as well as replicates, were considered as the “whole plot” factors, while harvest times (*n* = 2) and their interactions with cultivars and treatments were considered as “subplot” factors. All these effects were treated as fixed, while the random effects of cultivar × treatment × replicate interaction were used as the whole plot error. The partitioned F-tests were performed using the SLICE statement to examine the significance of treatments within cultivar × time interactions, followed by Tukey’s HDS test. The analysis of variance (ANOVA) was carried out using the MIXED procedure in SAS v. 9.4 [40].

## 3. Results

The field trial with two sweet pepper cultivars was conducted to test the effect of commercial calcite nanoparticles on sweet pepper yield, morphological, multispectral, physical, and chemical fruit properties. Fruit properties were analyzed by a combination of conventional and novel image-based techniques, and a list of all analyzed parameters is given as Appendix A. In addition, the analysis of variance (ANOVA) and the mean values with letters indicating significant differences for all analyzed properties are given as Appendix A.

### 3.1. Yield

Fruit yield (t ha^−1^) and the number of fruits per plant (Figure 2) were analyzed in the second harvest only (physiological maturity). A higher average fruit number per plant (12.8) was determined for Kurtovska kapija compared to Šorokšari (9.1). However, a higher average yield (42.8 t ha^−1^) was obtained for Šorokšari compared to Kurtovska kapija (32.2 t ha^−1^). Fruit number was not significantly affected by the treatments, whereas T3 significantly reduced fruit fresh weight, thus reducing the yield compared to all other treatments (T1, T2, and T4) for Šorokšari and compared to T4 treatment for Kurtovska kapija (Appendix A).

### 3.2. Chemical Properties of the Fruit

In both cultivars, higher sugar content, higher total organic acids’ fruit content (Figure 3A), and lower fruit pH were found at the second harvest (physiological maturity) compared to the first harvest (technological maturity). In addition, higher sugar content, higher total organic acids’ content, and lower pH was found in Kurtovska kapija compared to Šorokšari.

The treatments had no significant effect on the total organic acids’ content. On the other hand, at the first harvest, fruit sugar content was significantly increased in Kurtovska kapija in T4 (7.1%) compared to T3 (5.9%). In the second harvest time for Šorokšari, T3 increased the fruit pH (5.4) compared to the control (T1; 5.1) (Appendix A).

In both cultivars, a decrease in average fruit N, P, K, Mg, Zn, and Cu content and an increase in Mn content were found in the second harvest compared to the first harvest, while Ca content increased only in Šorokšari. In addition, Šorokšari had higher average fruit mineral content compared to Kurtovska kapija (Appendix A).

The most profound effect of treatments on fruit nutrient content was obtained for Šorokšari at the first harvest, where all treatments (T2, T3, and T4) significantly increased N, P, K, Mg, Fe, Zn, Mn, and Cu content compared to the control (T1) (Appendix A). However, there was no difference in Ca content among T1, T2, and T3, while the highest Ca content was found in T4 (0.18%) (Appendix A and Figure 4A). In the second harvest of Šorokšari, higher Ca, Fe, Zn, Mn, and Cu contents were found in all foliar treatments (T2, T3, and T4) compared to the control (T1). Hence, T2 significantly increased Mg (0.19%) and N (3.33%) content compared to T1 (0.16% Mg and 3.06% N) and T4 (0.15% Mg and 2.98% N). In Šorokšari, the highest Fe content (98.2 mg kg^−1^) was found in T4 at the first harvest, while the highest N (3.9%) and Mn (17.6 mg kg^−1^) contents were found in T3. In the second harvest, the highest N, P, Ca, Mg, and Fe contents were found in T2 treatment (Appendix A).

Unlike Šorokšari, foliar treatments did not have such a profound positive effect on fruit nutrient content in Kurtovska kapija. Moreover, compared to all treatments (T2, T3, and T4), the fruits of the control treatment (T1) had the highest N and Fe content at the first harvest and the highest Zn content at the second harvest. A significant positive effect of treatments on nutrient content in Kurtovska kapija fruits was found only for T3 treatment, which increased fruit Ca content (0.14%) (Figure 4A) at the first harvest and for T4, which increased P content (0.32%) at the second harvest (Appendix A).

### 3.3. Physical and Morphological Properties of the Fruit

Average fruit firmness and pericarp thickness (Figure 5) were higher in Šorokšari compared to Kurtovska kapija, and also in the second harvest compared to the first harvest. In the first harvest, significantly higher fruit firmness was found in T4 (22.1 kg cm^−2^) compared to T3 (17.00 kg cm^−2^) for Šorokšari. The opposite was found for Kurtovska kapija where the highest firmness was determined in T2 (21.7 kg cm^−2^), which was significantly higher than in T4 (15.6 kg cm^−2^).

In the second harvest, significant differences in fruit firmness and pericarp thickness were found only in Šorokšari, that is, the highest firmness was found at T3 (29.5 kg cm^−2^), followed by T4 (27.8 kg cm^−2^), while the lowest was found at T1 (20.6 kg cm^−2^). In addition, the highest pericarp thickness was determined at T2 (7.72 mm), while the lowest values were determined in T4 (6.41 mm) and T1 (6.44 mm).

Compared to Kurtovska kapija, the fruits of Šorokšari had a bigger average convex hull and width and a smaller average length, minimal circle radius, minimal circle area, and minimal rectangle area. All measured morphological parameters, except convex hull, increased in the second harvest for Šorokšari, while in Kurtovska kapija the convex hull, width, and minimal rectangle area increased, and the length, minimal circle radius, and minimal circle area decreased at the second harvest (Appendix A).

The effects of foliar treatments on fruit morphological parameters were detected only at the second harvest. In the second harvest in Šorokšari, T2 and T3 increased fruit length, minimal circle area, and minimal circle radius and decreased fruit width and convex hull compared to T4 and T1. The opposite effect was found in Kurtovska kapija, where T2 and T3 increased fruit width and convex hull compared to T4 and T1 in the second harvest (Appendix A).

### 3.4. Multispectral Properties of the Fruit

Significant differences between the two studied cultivars were found for all measured multispectral fruit properties. Compared to Šorokšari, higher average HUE, SAT, NIR, CHI, and ARI (Figure 6 and Figure 7) and lower average VAL, SpcGreen, and FarRed were found for Kurtovska kapija. Moreover, during fruit ripening (i.e., at the second compared to the first harvest), a decrease in HUE, SpcGreen, NIR, and CHI and an increase in SAT were found for both cultivars, while VAL and FarRed increased for Kurtovska kapija and decreased for Šorokšari (Appendix A).

The most profound effect on the studied multispectral properties was found at the second harvest for T2 and T3 treatments, with their contrasting effects on the studied cultivars. Specifically, in the second harvest, higher VAL, FarRed, and lower ARI and CHI in Kurtovska kapija were found in T2 and T3 compared to T1 and T4, whereas the opposite was found in Šorokšari (i.e., lower VAL and FarRed, and higher ARI and CHI). In addition, T3 increased SpcGreen (4662.7) compared to T1 (1445.3) and T4 (1292.4) in Kurtovska kapija (Appendix A).

## 4. Discussion

In evaluating the effect of calcite nanoparticles on the yield and on the multispectral, physical, morphological, and chemical properties of sweet pepper fruit, we used a combination of conventional and novel image-based nondestructive fruit quality analyses. To the best of our knowledge, there is no information in the literature on the effect of calcite nanoparticles in the form of CaCO_3_ on pepper fruit quality.

### 4.1. Yield

Lee et al. [41] suggested that the calcite application might have a positive effect on the fruit set. However, our results showed that the fruit number per plant was not affected by applied foliar treatments. In addition, the higher recommended concentration of calcite nanoparticles significantly reduced the fruit yield compared to all other treatments (T1, T2, and T4) for Šorokšari and compared to the T4 treatment for Kurtovska kapija (Figure 2).

Similar results were obtained by Tantawy et al. [42], who found that a lower concentration (0.5 g L^−1^) of 80.2% nano calcium carbonate had the greatest positive effect on tomato fruit yield (weight and number) compared to a higher concentration (1.0 g L^−1^). Moreover, Zaman et al. [43] found that the fruit weight of Kinnow mandarins was higher at 3% CaCO_3_ (166.03 g) than at 4% CaCO_3_ (152.13 g) with no significant differences. The reason for these observations might be related to the negative effects of calcite nanoparticles after application at high concentrations [44]. According to Zahedi et al. [11], nanoparticles have high penetration ability in plant tissues, and high concentrations of nanoparticles can negatively affect growth and development. 

### 4.2. Chemical Properties of the Fruit

The genetic background of the cultivar influences the physical and chemical fruit characteristics, and a higher average fruit mineral content was found in Šorokšari compared to Kurtovska kapija for all studied minerals. These results are consistent with those of previous studies [45,46,47]. Regardless of the cultivar, the mineral content in sweet pepper fruits followed an order of K > N > P > Ca > Mg, which showed the same trend as reported in the literature [26,48,49]. In addition, the following order of microelement content was determined: Fe > Zn > Cu > Mn, which agrees with the previously mentioned authors only for Fe and Zn but not for Cu and Mn (Appendix A). This can be explained by the fact that Ca interferes with the uptake of other elements such as Cu [50] and Mn [4]. Both calcite nanoparticle treatments (T2 and T3) as well as calcite-based foliar treatment (T4) had a more profound effect on the mineral content of Šorokšari and resulted in higher N, P, K, Mg, Fe, Zn, and Mn levels compared to the control (T1) at the first harvest and higher Ca, Fe, Zn, Mn, and Cu levels at the second harvest, which could also be positively affected by a favorable ratio between the precipitation and temperature during the vegetation period (Figure 1). While an increase in fruit mineral content obtained in T4 could be explained solely as a positive effect of foliar fertilizer, since the T2 and T3 treatments contained only calcite nanoparticles without other plant nutrients, the increased fruit mineral content found in the T2 and T3 treatments cannot be related to the chemical composition of the applied product. Rather, the increase is the result of changes in metabolism, mineral assimilation, and translocation within the plant [8,11,16,17]. For example, Azeez et al. [51] stated that Ca supports metabolic processes that affect the absorption of other minerals, and it can increase the uptake and translocation of different elements such as N, P, and K [4]. However, in contrast to the results found on Šorokšari, foliar treatments had either a negative (N, Fe, and Zn) or no effect on the microelement fruit content in Kurtovska kapija, and a positive effect was found only for fruit Ca content obtained in T3 at the first harvest and for fruit P content obtained in T4 at the second harvest. These results point not only to the variable effects of foliar calcite nanoparticle treatments but also of calcite-based foliar fertilization, on the assimilation of minerals in sweet pepper fruits of different cultivars. Similarly, Gulbagaca et al. [2] stated that nanoparticles can show both positive and negative effects on plant growth and development; they also noted that the effect of nanoparticles varies depending on the plant species and it is related to the composition, concentration, size, and the physical and chemical properties of nanoparticles.

In similar studies, foliar-applied CaCO_3_ in tomato [52] and grape [53] showed no effect on fruit pH, sugar, and total organic acids’ content. In contrast to these findings, in the present study, calcite-based foliar fertilizer (T4) applied to Kurtovska kapija showed a significant effect on the sugar content at first harvest (highest sugar content in T4), whereas high calcite nanoparticle treatment (T3) increased fruit pH at second harvest in Šorokšari. Results show that during ripening (from first to second harvest), both sugar and total organic acids’ content increased and pH decreased in pepper fruit. Although nonsignificant, in second harvest for Šorokšari, lower fruit sugar, lower total organic acids’ content, and higher pH were found for all calcite foliar treatments (T2, T3, and T4) compared to T1, indicating a slower ripening process in plants treated with calcite products. This could be explained by the effect of calcium on a reduced ethylene synthesis, which thus delayed fruit aging [54].

### 4.3. Physical and Morphological Properties of the Fruit

Since consumer perceives the external/visual fruit quality first, great attention should be given to the fruit size shape. Firmness is also a critical parameter for fruits [55], and declining firmness is the main reason for fruit quality loss [56]. Therefore, fruit quality can be improved if the size, firmness, and pericarp thickness are increased, and calcium is known to have great role in cell wall stability and therefore in the fruit development [57].

Results of this study showed that all calcite foliar treatments (T2, T3, and T4) increased fruit firmness, whereas calcite nanoparticle treatments (T2 and T3) increased the pericarp thickness for Šorokšari at second harvest. These results are probably related to the increased fruit calcium content, which was found at second harvest in Šorokšari fruits from these treatments. Calcium interacts with pectic acid in the cell wall, forming calcium pectate and improving maintenance of the cell wall structure. Moreover, it protects the cell wall from degrading enzymes that adversely affect firmness [58]. Similarly, calcium application has been shown to enhance the firmness of Jonathan [59], Golden Smoothee [60], and Braeburn [61] apples, and pear [62].

The effect of calcite nanoparticle treatments on fruit morphological parameters was evident at second harvest. Compared to T1 and T4, calcite nanoparticle treatments (T2 and T3) increased the fruit length, minimal circle area, and minimal circle radius, and they decreased the fruit width and convex hull in Šorokšari while increasing the fruit width and convex hull in Kurtovska kapija. Previous studies found that foliar calcite applications can increase the fruit size in tomato [52] and blueberry by 5% [63]. However, results of our study indicate a differential effect of calcite nanoparticle treatments on pepper fruit morphology with respect to their different genetic backgrounds. These results suggest that calcite nanoparticles interact with the fruit type (bell shaped or capia type), and this could be related to the Ca^2+^ role in cell wall stability [64], the effect of Ca^2+^ on hormonal balance in plants during growth and senescence [54], and its effect on lower hydrolyzing enzyme activities [65], which would be interesting to further investigate.

### 4.4. Multispectral Properties of the Fruit

The most obvious differences in the studied multispectral properties were obtained at second harvest, when calcite nanoparticle treatments (T2 and T3) increased value (VAL) and far red (FarRed) while decreasing the anthocyanin index (ARI) and chlorophyll index (CHI) compared to T1 and T4 in Kurtovska kapija; in Šorokšari, they decreased VAL and FarRed, and increased ARI and CHI. Higher fruit chlorophyll and anthocyanin content found in calcite nanoparticle treatments are probably related to the higher nutrient content—especially Ca and Fe—that was found under T2 and T3 treatments in Šorokšari fruits at second harvest, and their possible effect on fruit senescence. It was reported that exogenous calcium application could affect chlorophyll content, cell membrane fluidity, and respiration rates, which are important in the regulation of senescence [1]. In addition, Tantawy et al. [42] reported a positive effect of foliar calcium application on tomato chlorophyll content with the highest response found on nano calcium treatment (0.5 g L^−1^), followed by a nano calcium treatment of 1.0 g L^−1^, and then a chelated calcium treatment (3 g L^−1^). Besides calcium, iron has a central role in chlorophyll biosynthesis [64], and it improves anthocyanin stability by forming complexes with them [66].

## 5. Conclusions

In the present study, the two sweet pepper cultivars Šorokšari and Kurtovska kapija showed different responses to the applied treatments, i.e., calcite nanoparticles (at concentrations of 3% and 5%), calcite-based foliar fertilizer (positive control), and water (negative control), in almost all studied properties. In general, the application of calcite nanoparticles reduced the yield and increased fruit firmness. The treatments had a higher effect on the chemical properties of Šorokšari compared to those of Kurtovska kapija. The effects of treatments on fruit morphological and multispectral properties were different depending on the cultivar. It can be concluded that the application of calcite nanoparticles in sweet pepper could improve the chemical, physical, morphological, and multispectral fruit properties, but further studies under different environmental and soil conditions including different cultivars are required for sustainability of yield and fruit quality improvement.

## Figures and Tables

**Figure 1 biomolecules-11-00832-f001:**
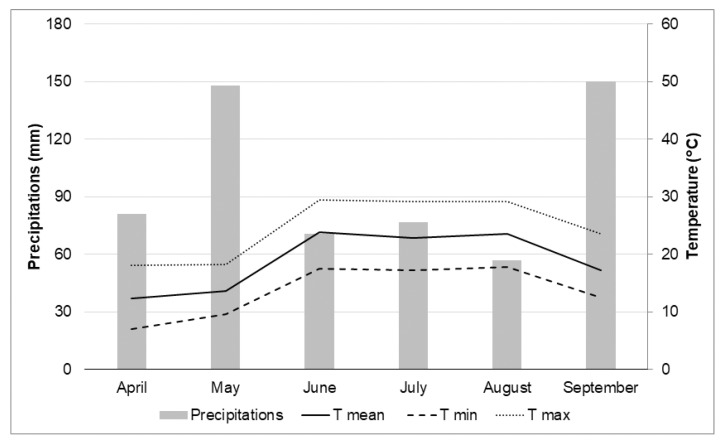
Walter climate diagram with the sum of precipitation and air temperature conditions during the experiment (Zagreb, Croatia). T min—minimum air temperature; T max—maximum air temperature; T mean—mean (average) air temperature.

**Figure 2 biomolecules-11-00832-f002:**
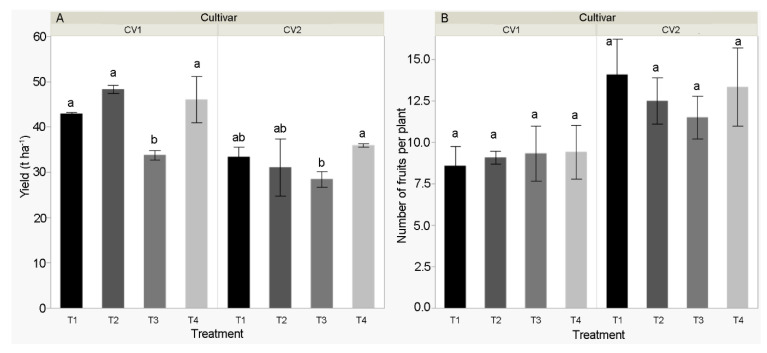
(**A**) Yield (t ha ^−1^) and (**B**) number of fruits per plant of sweet pepper cultivars CV1 (Šorokšari) and CV2 (Kurtovska kapija) fruits in H2 (second harvest at physiological maturity). The T1 (control), T2 (Eco Green, calcite nanoparticles at lower concentration), T3 (Eco Green, calcite nanoparticles at higher concentration), and T4 (Zeogreen+P, calcite-based foliar fertilizer) were applied three times during vegetation on sweet pepper plants. Post hoc comparisons of the means were performed using Tukey’s HSD test at *p* < 0.05; different letters indicate significant differences among treatments and cultivar.

**Figure 3 biomolecules-11-00832-f003:**
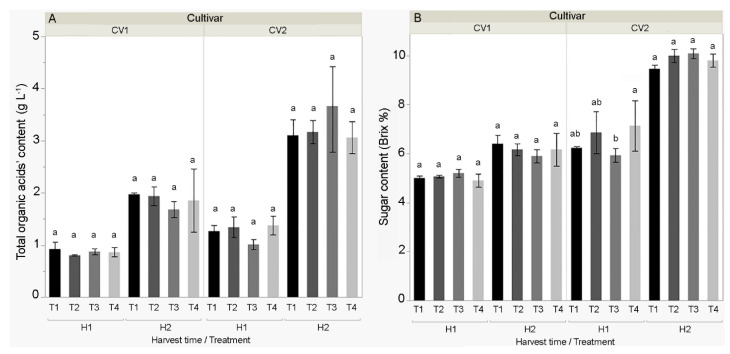
(**A**) Total organic acids’ content (g L^−1^) and (**B**) sugar content (% Brix) of sweet pepper cultivars CV1 (Šorokšari) and CV2 (Kurtovska kapija) fruits in H1 (first harvest at technological maturity) and H2 (second harvest at physiological maturity). The T1 (control), T2 (Eco Green, calcite nanoparticles at lower concentration), T3 (Eco Green, calcite nanoparticles at higher concentration), and T4 (Zeogreen+P, calcite-based foliar fertilizer) were applied three times during vegetation on sweet pepper plants. Post hoc comparisons of the means were performed using Tukey’s HSD test at *p* < 0.05; different letters indicate significant differences among treatments within each measurement time.

**Figure 4 biomolecules-11-00832-f004:**
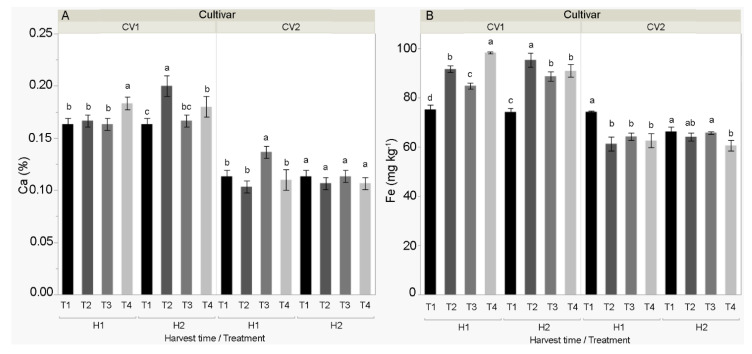
Selected minerals (**A**) calcium (Ca, %) and (**B**) iron (Fe, mg kg^−1^) of sweet pepper cultivars CV1 (Šorokšari) and CV2 (Kurtovska kapija) fruits in H1 (first harvest at technological maturity) and H2 (second harvest at physiological maturity). The T1 (control), T2 (Eco Green, calcite nanoparticles at lower concentration), T3 (Eco Green, calcite nanoparticles at higher concentration), and T4 (Zeogreen+P, calcite-based foliar fertilizer) were applied three times during vegetation on sweet pepper plants. Post hoc comparisons of the means were performed using Tukey’s HSD test at *p* < 0.05; different letters indicate significant differences among treatments within each measurement time.

**Figure 5 biomolecules-11-00832-f005:**
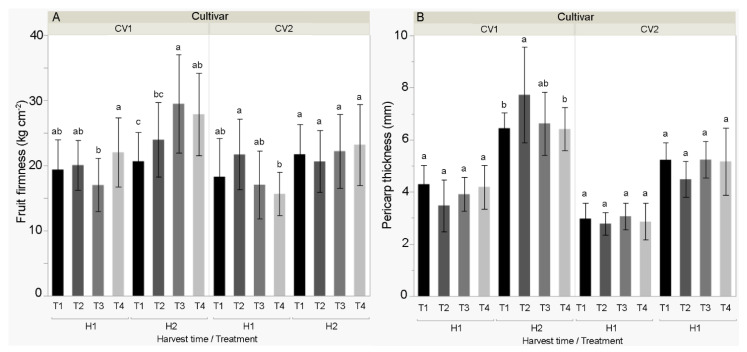
Selected physical fruit properties (**A**) fruit firmness (kg cm^−2^) and (**B**) pericarp thickness (mm) of sweet pepper cultivars CV1 (Šorokšari) and CV2 (Kurtovska kapija) fruits in H1 (first harvest at technological maturity) and H2 (second harvest at physiological maturity). The T1 (control), T2 (Eco Green, calcite nanoparticles at lower concentration), T3 (Eco Green, calcite nanoparticles at higher concentration) and T4 (Zeogreen+P, calcite-based foliar fertilizer) were applied three times during vegetation on sweet pepper plants. Post hoc comparisons of the means were performed using Tukey’s HSD test at *p* < 0.05; different letters indicate significant differences among treatments within each measurement time.

**Figure 6 biomolecules-11-00832-f006:**
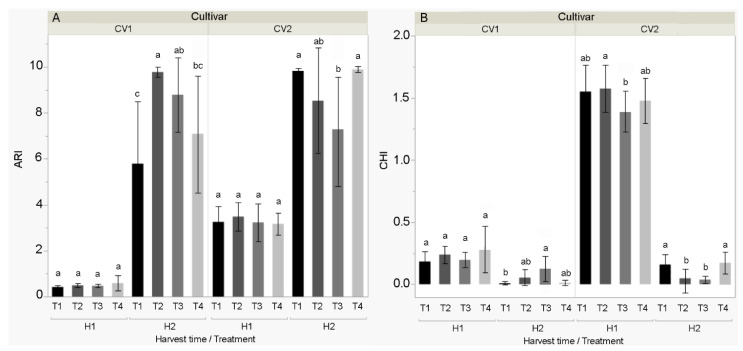
Selected multispectral fruit properties (**A**) anthocyanin index (ARI) and (**B**) chlorophyll index (CHI) of sweet pepper cultivars CV1 (Šorokšari) and CV2 (Kurtovska kapija) fruits in H1 (first harvest at technological maturity) and H2 (second harvest at physiological maturity). The T1 (control), T2 (Eco Green, calcite nanoparticles at lower concentration), T3 (Eco Green, calcite nanoparticles at higher concentration) and T4 (Zeogreen+P, calcite-based foliar fertilizer) were applied three times during vegetation on sweet pepper plants. Post hoc comparisons of the means were performed using Tukey’s HSD test at *p* < 0.05; different letters indicate significant differences among treatments within each measurement time.

**Figure 7 biomolecules-11-00832-f007:**
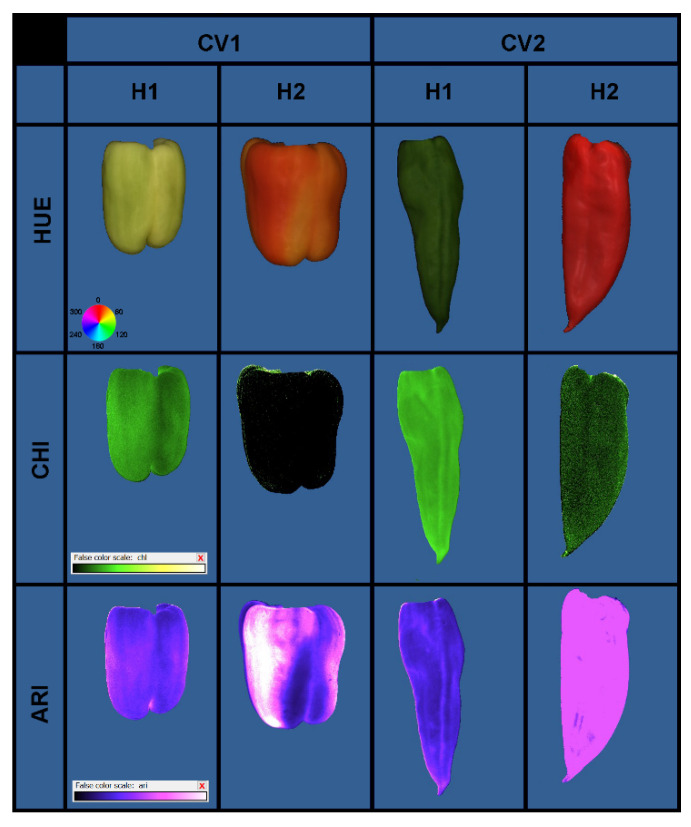
Color and pseudocolor images of hue (HUE), chlorophyll index (CHI), and anthocyanin index (ARI) of sweet pepper cultivars CV1 (Šorokšari) and CV2 (Kurtovska kapija) fruits at H1 (first harvest at technological maturity) and H2 (second harvest at physiological maturity).

**Table 1 biomolecules-11-00832-t001:** Chemical and physical soil properties at the experimental station.

pH_H2O_ ^a^	pH_KCl_ ^a^	%		mg 100 g^−1^		%		
		C_org_ ^b^	N ^c^	P_2_O_5_ ^d^	K_2_O ^d^	Sand ^e^	Silt ^e^	Clay ^e^
6.19	5.01	0.85	0.08	12.9	18.8	16.8	64.9	18.3

^a^ pH potentiometrically [32]; ^b^ Organic carbon content (C_org_) determination after dry combustion [33]; ^c^ Total nitrogen by the modified Kjeldahl method [34]; ^d^ Phosphorus and potassium by ammonium lactate method in accordance with Egner–Riehm–Domingo [35]; ^e^ Soil particle size distribution was determined by the pipette method with sieving and sedimentation [33].

**Table 2 biomolecules-11-00832-t002:** Treatments used in the experiment.

Abbreviation	Treatment (Product)	Product Calcite Content	Product Calcium Content	Applied Recommended Concentration (g L^−1^)
T1	Control	^−^		Water only
T2	Eco Green	50.80% CaCO_3_	20.32% Ca	3
T3	Eco Green			5
T4	Zeogreen+P	69.80% CaCO_3_	27.95% Ca	5

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
