# Peer review of "Multispectral Assessment of Sweet Pepper (Capsicum annuum L.) Fruit Quality Affected by Calcite Nanoparticles"

_biomolecules, 2021, doi:10.3390/biom11060832_

Round 1

Reviewer 1 Report

#General

The work developed by Vidak and collaborators, addresses carefully the effects of CaCO3 nanoparticles on the agronomical traits and multi-spectral properties of sweet pepper fruits. The methods seem to be well applied, and the results are interesting. In my opinion this work is timely within the current state of nano-enabled agriculture, and should be accepted for publication. However, I noticed few things that could/should be improved, which I list below.

Therefore, the authors should pay more effort to highlight the significance of this research in the abstract and conclusion sections and to show what gap they aim to fill.

#Methods

It needs characterization of the CaCO3 nanomaterials for this paper.

It would be beneficial to applying a multivariate statistic on these research data. It would help remarkably to the understanding of the cultivar’s effect when in contact with the CaCO3 nanoparticles.

#Results and discussion

Lines 240-241: In both cultivars the authors observe an increase in Mg content in the second harvest, but this increase are not linked with chlorophyll amount by CHI index (Fig 6b). Can the authors give more details about Mg role in fruit tissues? This result may be better discussed in discussion section.

Lines 409-411: This study used two cultivars, and suggest differential effect of calcite nanoparticles treatments on pepper fruit morphology with the respect of their different genetic background. NPs effects vs. phenotypic effects should be different from each other. Could the authors comment on this aspect?

Reviewer 2 Report

Major Comments

  • Abstract is too short (only 162 words and should be increased
  • Introduction- insufficient presentation of literature data; no justification of CaC03 NP utilization

Discuss and cite the works:

a)Buczkowska H., Michałojć Z., Konopińska J., Kowalik P. 2015. Content of macro- and microelements in sweet pepper fruits depending on foliar feeding  with calcium. J. Elem., 20(2): 261-272. DOI: 10.5601/jelem.2014.19.3.712

  1. b) Żurawik A., Jadczak D., Panayotov N., Żurawik P. (2020): Macro- and micronutrient content in selected cultivars of Capsicum annuum L. depending on fruit coloration. Plant Soil Environ., 66: 155-161.
  2. c) Kuo-Hsun Hua, Hsiang-Chuan Wang, Ren-Shih Chung, Ju-Chun Hsu Calcium carbonate nanoparticles can enhance plant nutrition and insect pest tolerance// Journal of Pesticide Science 2015,40(4):1-6; DOI: 10.1584/jpestics.D15-025
  • Materials and Methods –are poorly written, no details- in general it will be extremely difficult to repeat the work based on the data presented:

-no data on CaC03 NP size; antioxidants content (carotenoids, ascorbic acid, polyphenols, antioxidant activity

-no data on the concentration of Ca fertilizer and Ca NP/ Words: ‘low’, ‘high’ concentrations are insufficient

  • Results and discussion

- no data are presented on dry matter content and morphometry of fruits though the authors indicate that they have obtained these data and accordingly no discussion of these results; a lot of inaccuracies;

- Line 230 ‘the first harvest, fruit sugar content was significantly increased in 'Kurtovska kapija' in T4 

(7.1%) compared to T3 (5.9%)’- but according to Fig 3B no statistically significant differences exist

- Table 2- detailed information is necessary about NP concentrations used (higher or lower- is insufficient)

- lines 244 ‘the most profound effect of treatments on fruit nutrient content was obtained for 

'Šorokšari' at the first harvest, where all treatments (T2, T3 and T4) significantly increased 

N, P, K, Mg, Fe, Zn and Mn content compared to the control (T1)’- only Ca and Fe data are given in Figures (the same for line 356-359)

  • Conclusion- from the data presented it is difficult to understand the differences between NP and ordinary Ca-fertilizer
  • References not all references in the text are included in the reference list (see Lines 170, 172 ‘Gitelson et al., 2003, 2001')

Minor comments

1)Line 11 ‘first three authors contributed equally’- is it necessary to write this data here?

2) Lines 5-8 it is necessary to indicate in brackets the initials of the authors after e-mail addresses. E-mail of the corresponding author is printed twice

3)Line 15 ‘As modern sustainable vegetable production relies on methods that increase quality and yield values, such as foliar  fertilizers, biostimulants and plant growth regulators, as well as recent innovative techniques such  as nanotechnology, have an important role in improving agricultural products.’- don’t see the subject

4)Lines 83-85) ‘sweet pepper makes its products important as functional foods and has a positive impact on nutritional value  for  human consumption  and may play a role in reducing human microelement deficiencies – what elements???- reference

5) Line 110- what was the size of Ca NP?

6) Lines 119-120 details of Soil analysis

7) Line 138 ‘g L-1 of tartaric acid’- should be changed to : ‘g tartaric acid L-1’

8) Lines 144-148- references to the appropriate methods

9) Line 172) HUE- use  capital letters as everywhere

10) Lines 179-182 decipher Sat and VAL

11) Lines 170, 172 ‘Gitelson et al., 2003, 2001'- these references are absent in the reference list, use numbers while citing

12) Lines 240-243 ‘In both cultivars, a decrease in average fruit N, P, K, Mg, Zn and Cu content and an 

increase in  Mn content was found  in  the  second  harvest compared to the  first  harvest, 

while Ca content increased only in 'Šorokšari'. In addition, 'Šorokšari' had higher average 

fruit mineral content compared to 'Kurtovska kapija'.’ And the nest paragraph- add  (Figure 4)’

13) Line 278 ‘In addition, the  highest pericarp thickness  was determined  at  T2  (7.72  mm),  while the  lowest were  determinedin at T3 (6.41 mm) and T1 (6.44 mm).’- is there no mistake? Think that T3 should be replaced by T4 (see Figure 5B)   

14) Line 338- add (Figure 2)’

15) line ‘373 ‘  nanoparticles  can  showed’- change to ‘show’

16) use organic acids’ instead of ‘acids’ and ‘total acidity’ but not acid content

17) Lines 415-418 ‘The most obvious differences in studied multispectral  properties  were obtained at  second harvest, when  calcite nanoparticles treatments (T2 and T3) increased  VAL, FarRed  and  decreased  ARI and CHI compared to T1 and T4 in 'Kurtovska kapija', whereas  decreased VAL and FarRed, and increased  ARI and CHI  in 'Šorokšari'.’ Decipher abbreviations otherwise it is highly difficult to read the text

18) Line 418 higher chlorophyll  and anthocyanin content found in calcite  nanoparticles treatments’ change to ; ‘higher fruit chlorophyll   and anthocyanin content found in calcite  nanoparticles treatments’

19) Line 424 ‘In addition,  Tantawy et al.  [33]  reported  positive effect of foliar calcium application  on  chlorophyll content with highest response found on nano calcium treatment (0.5 g L-1) followed by the nano calcium treatment of 1.0 g L-1, and then the chelated calcium treatment (3 g L-1)-  what crop ???

Reviewer 3 Report

Multispectral Assessment of Sweet Pepper (Capsicum annuum L.) Fruits Quality Affected by Calcite Nanoparticles

Monika Vidak, Boris Lazarević, Marko Petek, Jerko Gunjača, Zlatko Šatović, Ivica Budor and Klaudija Carović-Stanko

A very well rounded and very well executed study on the effects of Calcite nanomaterials exposure on the growth, proliferation, agronomy, mineral content, and nutrients in two different cultivars of Capsicum annum L. Very sound experimental design and methodology and have been implemented well. Some minor corrections have been suggested underneath.

Line 105. On Figure 1, work on improving the picture quality of the graphic. It is an important graph for the results. Also incorporate the data in the discussion somewhere to support the findings.

Line 231: “……T3 increased fruit pH (5.4) compared to the control (T1; 5.1)”. Please point to the table or graph where this data is present. It could even be the supplementary information.

Line 240-243: One more time, please refer to the tables or figures that support this part of the results.

Line 244-246: Please direct the readers to the tables or figures where these results have been reported.

Line 253: Please direct the readers to the tabular data or figures. Please check throughout.

Line 264. Please check the spelling “technological/techonolgical”. Please check throughout.

Line 324: “…..(ARI) of of sweet pepper cultivars CV1………..”.

Line 351-355: “Regardless of the cultivar, the mineral content……..but not for Cu and Mn”. You could suggest a reason or a rationale, why.

Line 371: “These results points to the variable effects of foliar calcite…..”.

Line 373: “…..stated that nanoparticles can showed both positive and negative effects on …….”.

Line 404: “Effect of calcite nanoparticles treatments on fruit morphological parameters……”. Please check throughput for the same.

Line 421: “…….and their possible effects on fruit senescence”. Please check for sentence construction.

Round 2

Reviewer 2 Report

line 119} exclude repetition: 'vitamin C and ascorbic acid'

Author Response

Repetition is removed.